# Volumetric Modulated Arc Therapy for Radiosurgery of Brain Metastases: A Single-Center Study

José Manuel Sánchez-Villalobos [1,2,*], Alfredo Serna-Berna [3], Juan Salinas-Ramos [4], Pedro Pablo Escolar-Pérez [4], Marina Andreu-Gálvez [5,6], Emma Martínez-Alonso [2], José Antonio Pérez-Vicente [1] and Miguel Alcaraz [6,*]

1   Department of Neurology, University Hospital Complex of Cartagena, 30202 Cartagena, Murcia, Spain; josea.perez7@carm.es
2   Department of Cell Biology and Histology, School of Medicine, Regional Campus of International Excellence, "Campus Mare Nostrum", IMIB-Pascual Parrilla, University of Murcia, 30100 Murcia, Spain; emma@um.es
3   Department of Medical Physics and Radiation Protection, University Hospital Complex of Cartagena, 30202 Cartagena, Murcia, Spain; alfredo.serna@carm.es
4   Department of Radiation Oncology, University Hospital Complex of Cartagena, 30202 Cartagena, Murcia, Spain; juan.salinas@carm.es (J.S.-R.); pedrop.escolar@carm.es (P.P.E.-P.)
5   Department of Otorhinolaryngology, Head and Neck Surgery, Reina Sofia University Hospital, 30003 Murcia, Spain; marina.andreu@um.es
6   Department of Radiology and Physical Medicine, School of Medicine, Regional Campus of International Excellence, "Campus Mare Nostrum", IMIB-Pascual Parrilla, University of Murcia, 30100 Murcia, Spain
*   Correspondence: josemanuel.sanchez3@um.es (J.M.S.-V.); mab@um.es (M.A.)

**Abstract:** Whole-brain radiation therapy and stereotactic radiosurgery are two treatment modalities commonly utilized to treat brain metastases (BMs). The aim of this study is to retrospectively analyze the main radio-oncologic and clinical-demographic aspects of a cohort of BM patients treated with Volumetric Modulated Arc Therapy for radiosurgery (VMAT-RS). This is a cross-sectional observational design study with a retrospective review of the medical records of patients with brain metastases treated with VMAT-RS between 2012 and 2018. Clinical and demographic data, with special attention to sex, age, primary tumor, brain tumor-related epilepsy (BTRE), number and brain location of BMs, Karnofsky Performance Status (KPS), the updated DS-GPA prognostic index, and the survival estimated according to the Kaplan–Meier model from the date of radiosurgery, were analyzed. One hundred and twenty-one patients with 229 BMs were treated with VMAT-RS. Patients presented 1–4 BMs, which were treated with five non-coplanar VMAT arcs. Sixty-eight percent of the patients had lung cancer, and 35% of the BMs were in the frontal lobe. The proportion of local control was 88.5%. BTRE prevalence was 30.6%. The median survival time (MST) was 7.7 months. In the multivariate analysis of the Cox regression model, KPS $\geq$ 70 (HR$_{KPS < 70}$ = 2.59; $p$ = 0.001) and higher DS-GPA (HR$_{DS-GPAII}$ = 0.55, $p$ = 0.022; HR$_{DS-GPAIII-IV}$ = 0.38, $p$ = 0.006) were associated with improved survival. In the univariate analysis, primary tumor, age, and the presence of metastases in the posterior fossa (PFBMs) were also significant. In conclusion, the VMAT-RS is a technique with an overall survival rate comparable to other radiosurgery techniques. The median survival is significantly longer for those with higher KPS and DS-GPA. Other variables, such as the type of primary tumor, age, and PFBMs, could also influence survival, although further studies are needed.

**Keywords:** VMAT; stereotactic radiosurgery; brain tumor-related epilepsy; overall survival; brain metastasis; brain location



## 1. Introduction

Currently, brain metastases (BMs) represent the most frequent intracranial tumor, occurring in about 20–40% of cancer patients [1,2]. Improved diagnostic imaging techniques (e.g., magnetic resonance imaging), as well as more effective treatment regimens, have contributed to the increased incidence of BMs, making the therapeutic approach to BMs an

emerging challenge [3–6]. The clinical management of patients with BMs currently includes both systemic treatment (chemotherapy and/or targeted therapies, among others) and local treatment (neurosurgical and/or radiotherapeutic treatment). Whole-brain radiotherapy (WBRT) and stereotactic radiosurgery (SRS) are two treatment modalities commonly used to treat BMs. Traditionally, radiotherapeutic treatment has been carried out by WBRT. This involves the administration of a radiation dose to the entire brain parenchyma (although hippocampal-sparing WBRT is more dose-selective for different areas of the brain), usually in multiple treatment sessions. However, recent evidence has revealed the potential development of cognitive impairment [7,8]. In this context, SRS has become increasingly relevant. The radiosurgery technique involves the administration of a highly concentrated dose of radiation to the lesion with an extremely strong dose gradient in the surrounding area in order to minimize side effects on healthy brain tissue [9].

Volumetric modulated arc therapy (VMAT) is a radiotherapy technique that has been rapidly implemented in most cancer treatment settings due to its high efficiency compared to other intensity modulated radiotherapy (IMRT) techniques [10]. One of the strengths of VMAT radiosurgery compared to other radiosurgery techniques (such as gamma-knife radiosurgery) is that it does not require a stereotactic frame (frameless technique), which, together with its shorter procedure time, has facilitated its integration into radiation oncology departments [11]. In this context, neuro-oncological variables that may have a significant impact on patient survival should also be assessed. To date, some factors that have been shown to influence the survival of patients with BMs are the type and histology of the primary cancer, treatment of the primary cancer, and clinical characteristics such as age, size/number of BMs, Karnofsky performance score, and the effect of extracranial disease [12,13]. Another emerging factor to take into consideration that could have an impact on both survival and some clinical aspects of patients with BMs would be tumor-related epilepsy. Epilepsy related to tumor lesions is present in 20–35% of patients with BMs. Among the main risk factors described are metastases of melanoma and pulmonary origin, those with hemorrhagic components, and cortico-subcortical localization [14,15].

This work aims to answer some questions of interest in relation mainly to the clinical characteristics of patients treated with SRS as well as the variables with potential impact on patient survival. So, the objectives of the present study are to analyze: (a) the main clinical-demographic characteristics of a cohort of patients with BMs treated with VMAT-RS including the prevalence of epilepsy in different primary tumor types; (b) overall survival (OS) after radiosurgery treatment and the potential prognostic factors for survival; and (c) local control after treatment.

## 2. Materials and Methods

### 2.1. Study Design

This is a cross-sectional observational design study with a retrospective review of the medical records of the patients with brain metastases treated with volumetric modulated arc therapy for radiosurgery (VMAT-RS) between October 2012 and January 2018. Patients were treated in the Department of Radiation Oncology of the University Hospital Complex of Cartagena (Murcia, Spain). Patients undergoing both single fraction (SRS) and fractionated radiosurgery (fSRS) were included in the study [16,17]. The data collection and observational study were approved by the Ethics Committee of the University Hospital Complex of Cartagena (Murcia, Spain).

### 2.2. Study Cohort

Sex, age, primary tumor, presence of extracranial metastatic disease, number of total BMs at the time of radiosurgery treatment, prevalence of epilepsy related to brain metastases, local treatments before and/or after SRS treatment, seizures, Karnofsky Performance Status (KPS), and the median survival time (MST) (calculated from the date of radiosurgery to the date of death or the last clinical follow-up) were collected for each patient. The updated Graded Prognostic Assessment (GPA) was used for breast, lung (NSCLC ade-



nocarcinoma and NSCLC non-adenocarcinoma), renal cell carcinoma, melanoma, and gastrointestinal tumors [18]. For all other primary tumors, the GPA (2008) [19] was used. Finally, the main molecular biomarkers of the different types of primary tumors found in our study cohort are listed. Regarding breast cancer, we analyze, on the one hand, the hormone receptor (HR) status, which refers to the presence or absence of estrogen or progesterone receptor positivity, and, on the other hand, the HER2 (human epidermal growth factor receptor 2) status. For lung cancer, patients were grouped into small cell lung carcinoma, non-small cell lung carcinoma (NSCLC) adenocarcinoma, and NSCLC non-adenocarcinoma. Finally, in the case of melanoma patients, the presence of a BRAF mutation was analyzed.

For each of the BMs, the anatomical locations in the CNS were collected, being classified into frontal, parietal, temporal, occipital lobes, basal ganglia, brainstem, and cerebellum. The gross tumor volume (GTV), planned target volume (PTV), planned treatment dose, and dose fractionation scheme were also collected. The assessment of local control to treatment is performed by applying the mRECIST 1.1 [20] criteria individually for each BM, which is classified into one of the following categories: progressive disease, stable disease, partial response, or complete response. Finally, we analyzed the biological effective dose (BED) of the different treatment schemes for all treated patients. Equation (1) [21].

$$\text{BED} = nd\left[1 + \frac{d}{\alpha/\beta}\right],\tag{1}$$

Equation (1): the biological effective dose (BED), $d$ = dose administered in each treatment session, $n$ = the number of treatment sessions, $\alpha$ and $\beta$ are the radiosensitivity coefficients.

*2.3. Treatment Technique*

The planning and treatment protocols were similar to those carried out previously [9,11]. First, after obtaining the magnetic resonance image (gadolinium-enhanced 3D T1-weighted imaging, Gd T1WI), GTV is contoured. Subsequently, the PTV is generated with a margin of 2 mm (GTV-PTV) to consider inaccuracies in both GTV delimitation and patient positioning. At the time of radiosurgery, patients are immobilized using a thermoplastic mask and a frameless fixation system. All patients are treated using a VMAT technique consisting of 5 non-coplanar arcs and 6 MV X-rays, using optimized rotational delivery in order to avoid collision between gantry and patient while simultaneously minimizing treatment time. Our study included patients treated with single fractions as well as those undergoing fractionated treatment schedules (fSRS). Thus, while the planning aimed for an isodose at 100% of the prescribed dose to cover at least 99% of the PTV and 100% of the GTV for treatments through single-fraction SRS, patients treated with fSRS received a dose of 98% of the PTV and 98% of the GTV. The treatment planning optimization process was carried out using the Eclipse 10.0 software (Varian Medical System, Palo Alto, CA, USA) [21], and doses were calculated via the analytical anisotropic algorithm with a 1.2 mm grid. The treatment was administered using a Varian iX lineal accelerator equipped with a multi-leaf collimator (MLC) with a 5 mm central leaf width at the isocenter and a 10 mm leaf width in the outer field. Patient positioning was performed with Cone Beam CT (CBCT) assistance prior to the first arc and following the third arc delivery, with the couch situated in the neutral position. Regarding the treatment doses prescribed for each BM, in general, for patients who had previously received whole brain radiotherapy, the prescribed dose was 12 to 15 Gy, while for the rest, the prescribed dose ranged from 16 to 20 Gy. Figure 1 shows an example of treatment planning for a patient with three BMs treated with a single isocenter.

*2.4. Statistical Analysis*

Statistical analyses were performed using IBM SPSS version 25 (IBM, New York, NY, USA) for Mac. Survival curves were performed according to the Kaplan–Meier model. OS was computed from the first SRS treatment. The nonparametric log-rank test was performed to analyze differences in OS between patient subgroups. Univariate and multivariate Cox

proportional hazards models were used to identify predictors of OS. Pearson's chi-square test and Fischer's exact test were performed for the evaluation of categorical variables. Means were compared using independent samples and Student's *t*-tests. When more than two means were compared, a Kruskal–Wallis test was performed. Levene's test was performed to evaluate the homoscedasticity of the samples. Significant differences were reached when the P values were lower than 0.05 ($p < 0.05$).

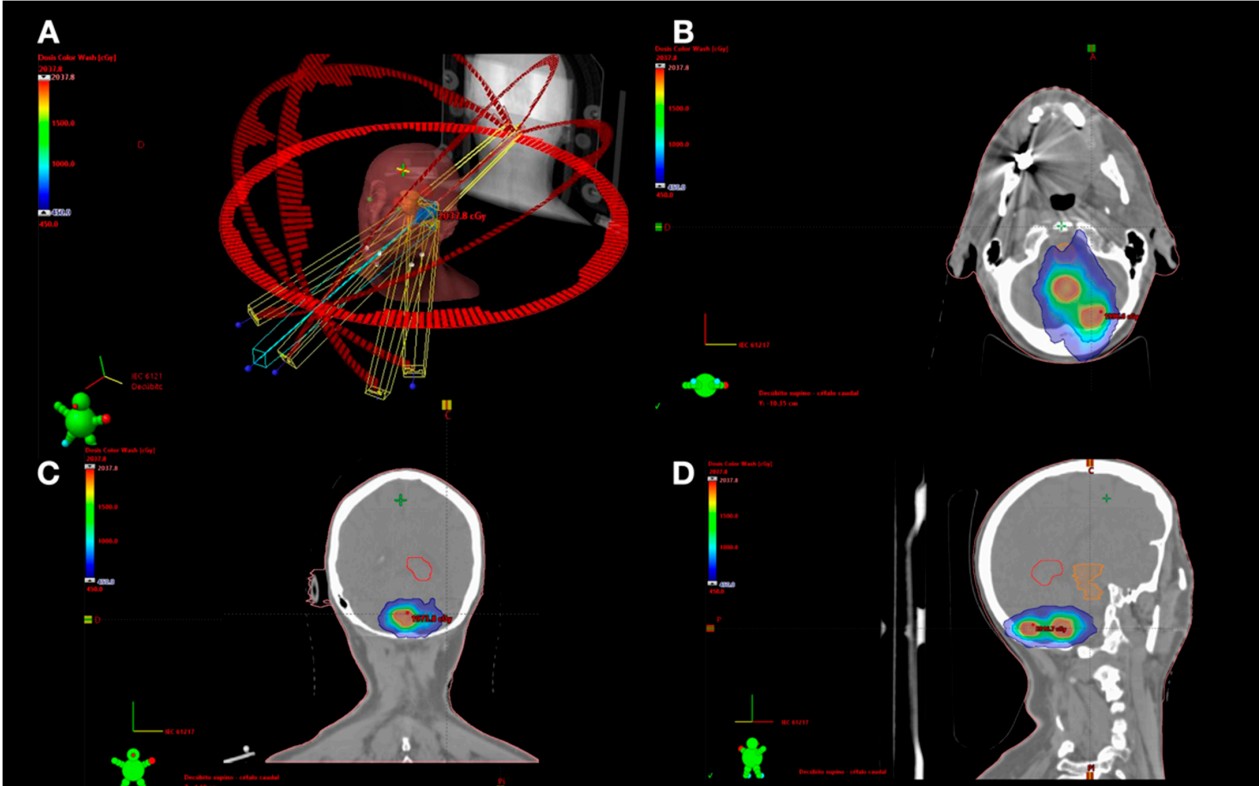

**Figure 1.** (**A**) Planning example of a patient with three brain metastases treated with 18 Gy. The images (**B–D**), represent the axial, coronal, and sagittal plane rotations, respectively. Red PTV outline, green 12 Gy isodose, light blue 9 Gy isodose, and dark blue 4.5 Gy. The 18 Gy isodose is not shown since it practically coincides with the PTV outline and would otherwise hinder viewing.

## 3. Results

### 3.1. Demographic Characteristics

A total of 123 patients with brain metastases included in radiosurgical treatment were collected. Of the total sample, two patients did not complete radiosurgical treatment due to clinical worsening, so they were excluded from the final study cohort. Therefore, a total of 121 patients with BMs treated with VMAT-RS (74 males and 47 females) were analyzed. Among patients with lung cancer, the prevalence of men was higher than that of women (74.4% men vs. 25.6% women, $p < 0.001$), while patients with breast cancer were all women. The mean age ($\pm$SD) of the patients was 62.7 $\pm$ 12.0 years (Figure 1A). The mean age of treated male patients was 64.3 $\pm$ 11.1, while that of females was 60.3 $\pm$ 13.1 years (Figure 2B–D), with no statistically significant differences ($p = 0.08$). Regarding the age distribution among the different types of primary tumors, on the one hand, those with gastrointestinal origin were the oldest (median $\pm$ IQR; 74 $\pm$ 17), followed by those with prostate origin (70.5). On the other hand, patients with BMs from breast cancer (52.5 $\pm$ 22) and melanoma (51) were the youngest ($p = 0.006$) (Figure 2E). Finally, the most frequent primary tumor was lung cancer (67.8%), followed by breast (13.2%), gastrointestinal (7.4%), genitourinary (5.0%), melanoma (2.5%), prostate (1.7%), and the remainder (2.5%) (Figure 2F, Table 1).

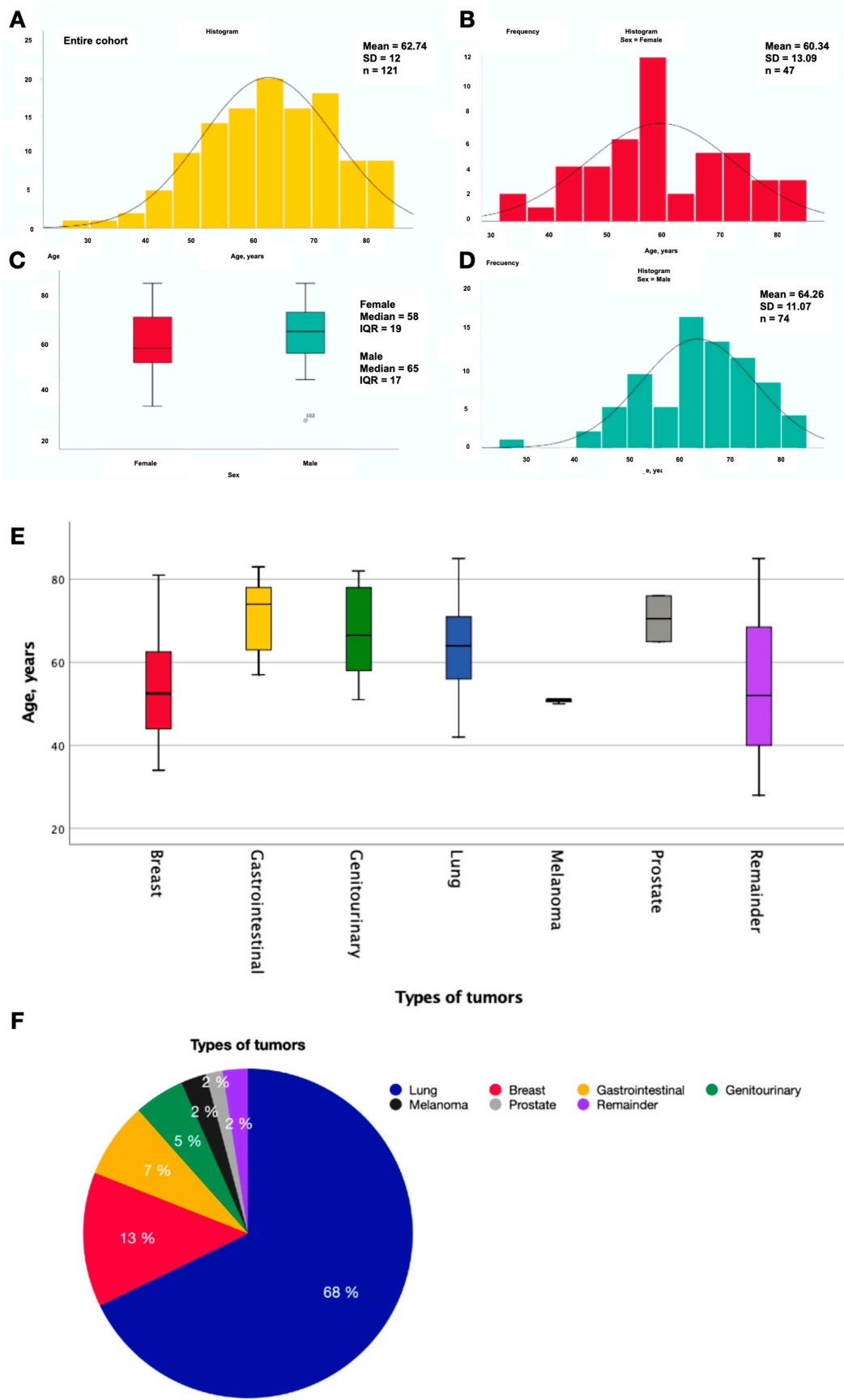

**Figure 2.** Histogram showing the age distribution of patients with brain metastases treated with VMAT-RS: (**A**) Complete cohort, (**B**) female cohort, and (**D**) male cohort. (**C**) Boxplot of the comparative

age variables between male and female subgroups. (**E**) Boxplot of comparative age variables between different types of primary tumors. (**F**) Sector diagram showing the relative frequencies of primary tumors in patients with BMs treated with VMAT-RS. Lung (67.8%, *n* = 82), breast (13.2%, *n* = 16), gastrointestinal (7.4%, *n* = 9), genitourinary (5.0%, *n* = 6), melanoma (2.5%, *n* = 3), prostate (1.7%, *n* = 2), and the remainder (2.4%, *n* = 3).

**Table 1.** Clinical characteristics of patients treated by SRS/fSRS.

| | | | |
|---|---|---|---|
| Age, years | Median (IQR) | 63 | (17) |
| Sex, *n* (%) | Female | 47 | (38.8) |
| | Male | 74 | (61.1) |
| Primary tumor, *n* (%) | Total cohort | 121 | (100) |
| Lung | Total lung cohort | 82 | (67.8) |
| | NSCLC adenocarcinoma | 56 | (46.3) |
| | NSCLC Non adenocarcinoma | 18 | (14.9) |
| | SCLC | 8 | (6.6) |
| Breast | Total breast cohort | 16 | (13.2) |
| | HER2+/HR+ | 7 | (5.8) |
| | HER2+/HR- | 3 | (2.5) |
| | HER2−/HR+ | 2 | (1.7) |
| | HER2−/HR− | 3 | (2.5) |
| | Sarcoma | 1 | (0.8) |
| Gastrointestinal | | 8 | (6.6) |
| Genitourinary | | 7 | (5.8) |
| Melanoma | Total | 3 | (2.5) |
| | BRAF negative/unknown | 1 | (0.8) |
| | BRAF positive | 2 | (1.7) |
| Remainder | Prostate | 2 | (1.7) |
| | Choriocarcinoma | 1 | (0.8) |
| | Oropharynx | 1 | (0.8) |
| | Unknown | 1 | (0.8) |
| KPS score, *n* (%) | 100–90 | 26 | (21.5) |
| | 80–70 | 53 | (43.8) |
| | <70 | 42 | (34.7) |
| DS-GPA Class, *n* (%) | 0–1 | 38 | (31.4) |
| | 1.5–2.0 | 43 | (35.5) |
| | 2.5–3.0 | 32 | (26.4) |
| | 3.5–4.0 | 8 | (6.6) |
| Epilepsy related to BMs, *n* (%) | Yes | 37 | (30.6) |
| | No | 84 | (69.4) |
| Overall survival, months | Median (IQR) | 7.72 | (0.905) |
| Extracranial metastases, *n* (%) | Yes | 70 | (57.9) |
| | No | 51 | (42.1) |
| Radiosurgery treatment, *n* (%) | Total BMs treated | 229 | (100) |
| | SRS | 206 | (90) |
| | Fractionated SRS (fSRS) | 23 | (10) |

**Table 1.** *Cont.*

| | | | |
|---|---|---|---|
| Previous treatment, *n* (%) | None | 88 | (72.7) |
| | Surgery | 3 | (2.5) |
| | WBRT | 27 | (22.3) |
| | Prophylactic WBRT | 3 | (2.5) |
| Posterior treatment, *n* (%) | None | 84 | (69.4) |
| | WBRT | 15 | (12.4) |
| | Single or fractionated SRS | 14 | (11.6) |
| | SRS + WBRT | 7 | (5.8) |
| | Surgery | 1 | (0.8) |
| PFBMs patients, *n* (%) | Patients with 1 or more PFBMs | 29 | (24) |
| | Patients without PFBMs | 92 | (76) |
| BMs treated with SRS (First treatment) | Median (IQR) | 1.0 | (1.0) |
| | Mean (SD) | 1.7 | (0.96) |
| Prescription dose BMs, Gy | SRS, median (IQR) | 18 | (0.8) |
| | fSRS, median (IQR) | 30 | (0.5) |
| Gross tumor volume, cc (GTV) | SRS, median (IQR) | 0.8 | (2) |
| | fSRS, median (IQR) | 4.1 | (10) |
| Planning target volume, cc (PTV) | SRS, median (IQR) | 2.5 | (5) |
| | fSRS, median (IQR) | 9.3 | (17) |
| Cumulative tumor volume, cc ($\Sigma$GTV) | Median (IQR) | 3.2 | (6.0) |
| $BED_{10\text{-}LQ}$, Gy | SRS, mean (SD) | 49.37 | (10.07) |
| | fSRS, mean (SD) | 48.1 | (2.82) |

$BED_{10\text{-}LQ}$, biologically effective dose for an alpha/beta ratio of 10, linear-quadratic model; BMs, brain metastasis; DS-GPA, diagnosis-specific graded prognostic assessment; HER2, human epidermal growth factor receptor 2; HR, hormone receptor; IQR, interquartile range; KPS, Karnofsky Performance Status; NSCLC, non-small cell lung cancer; PFBMs, posterior fossa brain metastasis; SCLC, small cell lung cancer; SRS, stereotactic radiosurgery.

### 3.2. Brain Metastases

A total of 229 BMs treated with VMAT-RS were collected from the entire cohort of 121 patients. Of the total BMs treated, 206 were treated as a single fraction (SRS), while 23 were treated fractionally (fSRS). The median dose prescribed for those patients treated with SRS was 18 Gy (range 12–20 Gy), while for those treated with fSRS it was 30 Gy (range 30–35 Gy) administered in 5–6 treatment fractions. The median accumulated tumor volume of the treated BMs was 3.25 cc (IQR 6.13).

Regarding the distribution of BMs in the CNS, the prevalence by topographic regions was as follows (from most to least frequent): (a) frontal lobe (*n* = 80; 34.9%), (b) parietal lobe (*n* = 45; 19.7%), (c) cerebellum (*n* = 33; 14.4%), (d) temporal lobe (*n* = 25; 10.9%), (e) occipital lobe (*n* = 23; 10%), (f) basal ganglia (*n* = 5; 2.2%), and (g) remainder (*n* = 18; 7.9%) (Figure 3). The latter group is made up of: brain stem (*n* = 4; 1.7%), insula (*n* = 3; 1.3%), sulcus of Rolando (*n* = 2; 0.9%), extra-axial (*n* = 2; 0.9%), thalamus (*n* = 1; 0.4%), subthalamic nucleus (*n* = 1; 0.4%), amygdala (*n* = 1; 0.4%), choroid (*n* = 1; 0.4%), hypophysis (*n* = 1; 0.4%), ventricular atrium (*n* = 1; 0.4%), and fornix (*n* = 1; 0.4%).

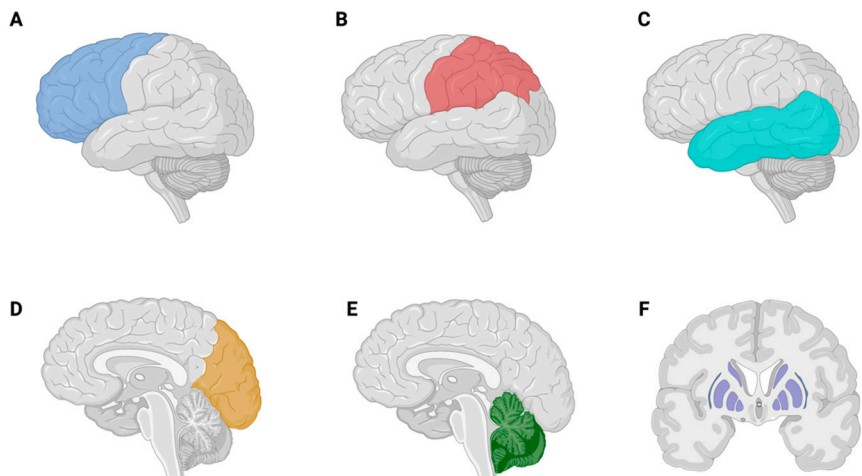

**Figure 3.** Distribution of brain metastases in the central nervous system. (**A**) Frontal lobe (*n* = 80; 34.9%), (**B**) parietal lobe (*n* = 45; 19.7%), (**C**) temporal lobe (*n* = 25; 10.9%), (**D**) occipital lobe (*n* = 23; 10%), (**E**) cerebellum (*n* = 33; 14.4%), (**F**) basal ganglia (*n* = 5; 2.2%), and remainder (*n* = 18; 7.9%).

Regarding the prevalence of posterior fossa BMs (PFBMs) in those according to the primary tumor (defined as those patients with at least one BM in the brainstem and/or cerebellum), it was found that patients with BMs of breast cancer were those with the highest prevalence (50%), while in those of pulmonary origin it was lower (23.2%).

### 3.3. Brain Tumor-Related Epilepsy

Of the entire sample, 37 patients developed epilepsy related to brain metastases (30.6%). Patients with a primary lung tumor accounted for 51.35% of all BTRE patients, followed by those with breast cancer (18.92%). In contrast, patients with melanoma had the highest prevalence of epilepsy within a tumor subtype (66.67%), although no statistically significant differences were found (*p* = 0.084) (Figure 4). We also found no statistically significant differences in the prevalence of epilepsy according to sex (*p* = 0.061) or age (*p* = 0.753).

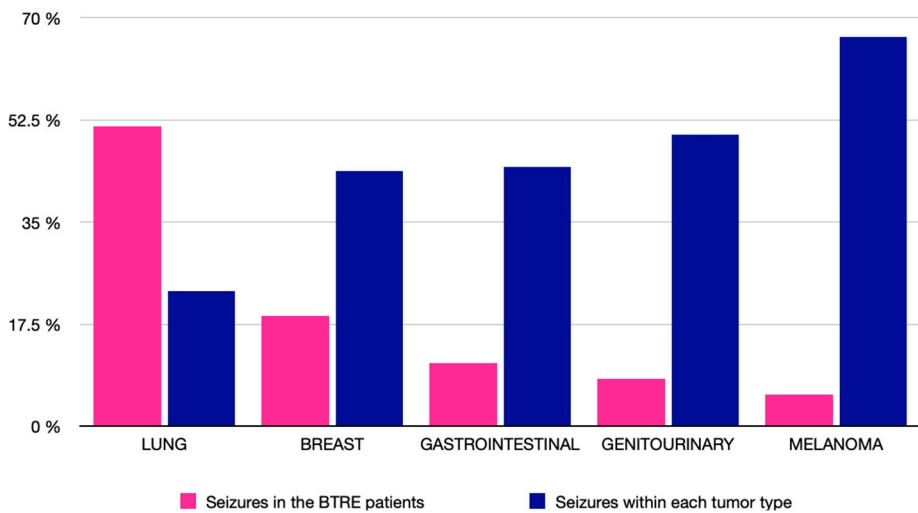

**Figure 4.** Prevalence of tumor-related epilepsy (BTRE). Prevalence of tumor types in the overall BTRE cohort (pink bar graph): lung (51.35%), breast (18.92%), gastrointestinal (10.81%), genitourinary (8.11%), and melanoma (5.41%). Prevalence of BTRE within each tumor subtype (blue bar graph): melanoma (66.67%), genitourinary (50%), gastrointestinal (44.4%), breast (43.75%), and lung (23.17%).

### 3.4. Survival Analysis

Survival analysis according to the Kaplan-Meier model is shown in Figure 5. The median survival time (MST) was 7.72 months (95% confidence interval: 5.9–9.49 months). OS rates were 59.5% at 6 months, 35.5% at 12 months, and 16.5% at 24 months. The log-rank test results of the Kaplan–Meier model for all factors are shown in Figure 5. Significant differences in survival were found for the following variables: (a) KPS; patients with KPS $\geq$ 70 had a median survival time (MST) of 12.06 (95% CI, 7.3–16.8), as opposed to those with KPS < 70 who had an MST of 2.2 (95% CI, 1.12–3.28) ($p < 0.001$); (b) DS-GPA Class; patients were divided into four groups: Class I (DS-GPA 0–1.0; $n = 38$), Class II (DS-GPA 1.5–2.0; $n = 43$), Class III (DS-GPA 2.5–3.0; $n = 32$), and Class IV (DS-GPA 3.5–4.0; $n = 8$). Due to the smaller sample size of the last two groups, for survival analyses, they were grouped into one group (Class III–IV, $n = 40$). Thus, the MSTs of the patients grouped according to DS-GPA were Class I at 2.56 months (95% CI 0.67–4.44), Class II at 7.75 months (95% CI 5.55–9.94), and Class III–IV at 14.09 months (95% CI 9.15–19.03) ($p < 0.001$). (c) Age; the MST for patients <65 years was 8.58 months (95% CI 4.3–12.86), while those $\geq$65 years presented a MST of 6.28 months (95% CI 2.95–9.6); (d) presence of PFBMs; the MST of the patients with the presence of PFBM was 14.39 (95% CI 5.09–23.69; $n = 29$), while that of those without was 6.44 (95% CI 4.65–8.24; $n = 92$) ($p = 0.029$); (e) the MST of patients with BMs of breast cancer origin was 19.42 months (95% CI 16.52–22.32), while that of those with lung cancer was 7.52 (95% CI 5.75–9.3) ($p = 0.015$). In contrast, no statistically significant differences were found between median survival times regarding seizures ($p = 0.96$) and the presence of extracranial BMs ($p = 0.28$). Finally, univariate and multivariate Cox regressions were performed (Table 2). In the univariate analysis, significant differences were found in the hazard ratios of the following variables: KPS ($HR_{KPS < 70} = 4.15$; $p < 0.001$), DS-GPA ($HR_{DS-GPAII} = 0.58$, $p = 0.018$; $HR_{DS-GPAIII-IV} = 0.32$, $p < 0.001$), age ($HR_{\geq 65} = 1.6$; $p = 0.014$), PFBMs ($HR = 0.62$; $p = 0.03$), and primary tumor type breast/lung ($HR_{breast} = 0.51$; $p = 0.017$), no significant differences were found between BM-related epilepsy ($p = 0.96$), cumulative tumor volume ($p = 0.19$), and ECM ($p = 0.29$). In multivariate analysis, only KPS ($HR_{KPS < 70} = 2.59$; $p = 0.001$) and DS-GPA ($HR_{DS-GPAII} = 0.55$, $p = 0.022$; $HR_{DS-GPAIII-IV} = 0.38$, $p = 0.006$) were statistically significant.

### 3.5. Local Control

Local response was analyzed individually for each BM. For local response assessment, the mRECIST 1.1 criteria for each BM were used [20]. Surgical bed irradiation was excluded. Neuroimaging monitoring was performed at a mean time of 2.9 months (SD: 1.4 months). In 92% of cases, it was performed by MRI, and in 8%, by cranial CT. In 24.5% of cases, neuroimaging control was not performed. In the remaining cases, the proportion of local control was 88.5%. Each BM individually met the criteria of: PD 11.5%, SD 43.9%, PR 29.3%, and CR 15.3%. There were no statistically significant differences in the proportion of local control between the single fraction (88.7%) and hypofractionated (86.7%) treatment modalities ($p = 0.971$).

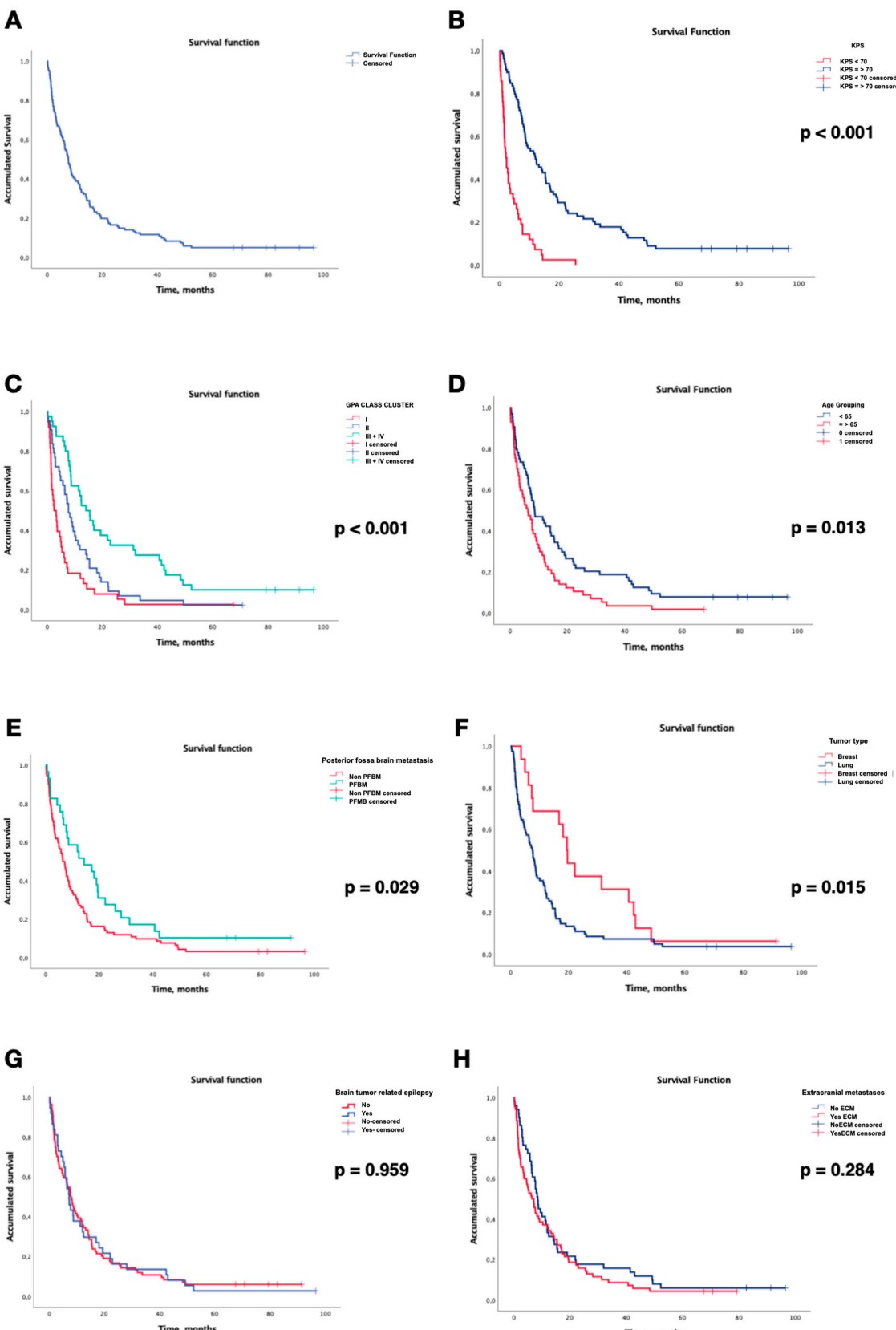

**Figure 5.** Kaplan–Meier curves of patients treated with SRS/fSRS for: (**A**) overall probability of survival. Additionally, survival according to (**B**) KPS (cut-off 70), (**C**) DS-GPA class, (**D**) age grouping (cut-off 65 years), (**E**) presence of posterior fossa brain metastasis, (**F**) tumor type (lung vs. breast), (**G**) brain tumor-related epilepsy, and (**H**) presence of extracranial metastasis.

**Table 2.** Univariate and multivariate analysis of prognostic factors.

| | Hazard Ratio (HR) | 95% CI | *p*-Value |
|---|---|---|---|
| **Univariate** | | | |
| KPS | | | |
| KPS ≥ 70 | 1 | | |
| KPS < 70 | 4.153 | 2.728–6.322 | <0.001 |
| DS-GPA Class | | | |
| Class I (0–1.0) | 1 | | |
| Class II (1.5–2.0) | 0.583 | 0.373–0.911 | 0.018 |
| Class III–IV (2.5–4.0) | 0.319 | 0.199–0.511 | <0.001 |
| Age | | | |
| <65 years | 1 | | |
| ≥65 years | 1.596 | 1.101–2.316 | 0.014 |
| Posterior fossa BMs | | | |
| No | 1 | | |
| Yes | 0.616 | 0.396–0.956 | 0.031 |
| Primary tumor type | | | |
| Lung | 1 | | |
| Breast | 0.505 | 0.289–0.885 | 0.017 |
| Epilepsy-related brain tumor | | | |
| Yes | 1.010 | 0.681–1.499 | 0.959 |
| No | 1 | | |
| Cumulative brain tumor | | | |
| ΣGTV ≥ 5 cc | 1.292 | 0.878–1.900 | 0.194 |
| ΣGTV < 5 cc | 1 | | |
| Extracranial metastases | | | |
| Yes | 1.225 | 0.844–1.778 | 0.286 |
| No | 1 | | |
| **Multivariate** | | | |
| KPS | | | |
| KPS ≥ 70 | 1 | | |
| KPS < 70 | 2.593 | 1.484–4.532 | 0.001 |
| DS-GPA Class | | | |
| Class I (0–1.0) | 1 | | |
| Class II (1.5–2.0) | 0.548 | 0.328–0.915 | 0.022 |
| Class III–IV (2.5–4.0) | 0.378 | 0.189–0.756 | 0.006 |
| Age | | | |
| <65 years | 1 | | |
| ≥65 years | 0.714 | 0.433–1.178 | 0.187 |
| Posterior fossa BMs | | | |
| No | 1 | | |
| Yes | 0.646 | 0.378–1.102 | 0.109 |
| Primary tumor type | | | |
| Lung | 1 | | |
| Breast | 0.731 | 0.403–1.327 | 0.303 |

BMs, brain metastasis; CI, confidence interval; DS-GPA, diagnosis-specific graded prognostic assessment; GTV, gross tumor volume; KPS, Karnofsky Performance Status.

## 4. Discussion

VMAT radiosurgery for the treatment of BMs is increasingly utilized in radiotherapy treatments due to its high level of efficiency compared to other intensity-modulated radiotherapy techniques [22]. Several previous studies analyzed dosimetric parameters

and procedure times, showing how the use of non-coplanar VMAT arcs presents high levels of dose conformality with low levels of exposure of healthy brain parenchyma [23] as well as reduced procedure times [9,11]. Another advantage shown by SRS over conventional WBRT is the lower neurocognitive impact on treated patients [7,8], an aspect of growing interest given the increased life expectancy of patients. However, at present, many questions remain to be resolved regarding survival, such as the potential impact of the location of BMs, the tumor burden, or the presence of tumor-related epilepsy, among others. This is one of the first studies to analyze holistically a cohort of patients with BMs treated with radiosurgery using volumetric modulated arc therapy, trying to offer a global radiography of the neuro-oncologic patient by analyzing both the main radio-oncologic and clinicodemographic variables.

Firstly, the most frequently found primary tumor in this cohort was lung cancer (67.8%), followed by breast cancer (13.2%), similar to previously reported [24]. Regarding sex distribution, the prevalence of men was higher than women in the cohort, also partly due to the higher prevalence of the male sex in the cohort of lung cancer patients (74.4% versus 25.6%, $p < 0.001$). Our study also showed that patients with breast cancer and melanoma were younger than those with gastrointestinal cancer or prostate tumors ($p = 0.006$). Regarding the distribution of the BMs, the frontal lobe (34.9% of the BMs) was the main location of the BMs, similar to previous works [25]. In contrast, only 16.1% of the BMs were in structures of the posterior fossa of the CNS (brainstem and/or cerebellum), with breast cancer metastases being the most frequent in this latter location. Several studies have found that the cerebellum is the predominant site of metastases in breast cancer patients [11,26–29]. Although the "seed and soil" theory has traditionally been used to explain localization in specific areas of the brain [30], other possible explanations have now been proposed to explain the preferential involvement of the cerebellum by breast cancer BMs that include both anatomical and hemodynamic aspects [29]. Therefore, this differential CNS distribution according to primary tumor could be of importance for the planning of treatment schemes in the future.

Secondly, the potential impact on the survival of brain tumor-related epilepsy patients treated with radiosurgery remains *terra incognita* at present. The prevalence of seizures reported in previous studies ranged from 20 to 35%, being more frequent in patients with BMs of lung cancer and melanoma. Other major factors known to increase the risk of seizures would be: supratentorial and cortico/subcortical junction localization and the presence of hemorrhagic components in BMs [14,31–35]. In our study, the prevalence was 30.6%, with the most frequent etiology in the overall BTRE cohort being of pulmonary origin (51.35%). Patients with melanoma BMs developed seizures most frequently (66.67%), although these results should be taken with caution given the small sample size of this subgroup of patients. Some demographic aspects have been scarcely analyzed in previous studies, and there is no clear consensus on the results obtained. Thus, while Puri et al. (2020) [36] et al. reported that age is the only variable that correlates negatively with the occurrence of pre- and postoperative seizures, other authors such as Witteler et al. (2020) [37] and Maschio et al. (2022) [38] did not observe any significant correlation between age and seizure risk. In our study, no statistically significant differences were found with respect to either sex ($p = 0.061$) or age ($p = 0.753$). Finally, another aspect of special interest is the therapeutic management of epileptic seizures. Its importance lies in the impact that the appropriate choice of antiseizure medication has both on seizure control and on the potential impact on the neurocognitive sphere and the quality of life of patients. However, given its complexity, this topic was addressed in particular elsewhere [14,15].

Thirdly, for the evaluation of the local response of BMs to radiosurgery treatment, the mRECIST criteria [20,39] were used, which represent an institutional modification of the RECIST 1.1 criteria [40]. One of the main differences between the two criteria focuses on the definition of measurable lesions [20]. Thus, with the mRECIST criteria, metastatic lesions with a minimum diameter of 5 mm are included instead of 10 mm, as postulated by the RECIST 1.1 criteria [40]. This increases the potential set of BMs studied and may

be more sensitive for detection of local disease progression. This is because it does not require a minimum absolute increase of 5 mm but only an increase $\geq 20\%$ in the sum of the longest diameters compared to the nadir value [20]. In our work, the mRECIST 1.1 criteria were applied individually to each BM, obtaining a local control percentage of 88.5% at a mean of 2.9 months from the date of radiosurgery (SD 1.4 months). This proportion of local control is comparable to the one obtained in our previous study, although now with a larger sample size [11]. In addition to the RECIST criteria, there are other criteria for assessing response to treatment, such as the Macdonald and BM-RANO criteria [41]. Some recent studies have observed a high degree of concordance between some of these criteria (mRECIST, RECIST 1.1, and BM-RANO) [20].

Finally, the MST from the date of radiosurgery was 7.7 months, which is comparable to the overall result obtained in the RTOG 9508 trial. These findings are also comparable to other previous studies that analyzed the survival of BM patients treated with radiosurgery, although some variability can be found in the literature: Bashir et al. (2014) (MST 8 months) [42], Serna et al. (2015) (7.2 months) [9], Kim et al. (2021) (8.2 months) [43], Park et al. (2021) (9.3 months) [44], or Mangesius et al. (2021) (11 months) [45]. In the multivariate Cox regression model, two variables were statistically significant: KPS and the updated DS-GPA. Regarding KPS, our results showed that a KPS < 70 at the time of radiosurgical treatment represents an increase in mortality of approximately 4 and 2.5 times, according to the univariate ($p < 0.001$) and multivariate ($p = 0.001$) analyses, respectively. This is in relation to previous studies, where using other radiosurgery techniques, KPS was shown to be an outstanding prognostic factor for survival [42,46]. Regarding DS-GPA, our results showed a reduction in the mortality of patients with DS-GPA Class II of 42% ($p = 0.018$) and of DS-GPA Class III–IV patients of almost 70% ($p < 0.001$) with respect to DS-GPA Class I patients. Sperduto et al. (2020) [18] in a multi-institutional database of 6.984 patients found significant differences between the DS-GPA groups described, with DS-GPA Class 1 (5 months), Class II (11 months), Class III (20 months), and Class IV (33 months) [18]. In our study, the estimated median survivals relative to each subgroup would be Class I (3 months, $n = 38$), Class II (8 months, $n = 43$), and Class III–IV (14 months, $n = 40$). In the latter case, given the low sample size of Class IV (23 months, $n = 8$), it was decided to pool Class III. One of the possible explanations for the lower MST in our cohort, in addition to variations due to the smaller sample size, is that in this study we used the date of death of the patients to estimate survival or, in those cases in which we do not have it, the date of the last clinical follow-up, and perhaps the latter may contribute to partially underestimating survival. On the other hand, in univariate analysis, other factors such as age, primary tumor (breast vs. lung), and the presence of PFBM were significantly related to survival. Regarding age, previous studies have shown an inverse correlation between age and survival in patients with BMs [18,19]. In our study, an increased mortality risk of 60% (HR 1.6, $p = 0.014$) was obtained for the subgroup of patients older than 65 years. The presence of posterior fossa metastasis as well as breast cancer as the primary tumor was statistically significant in the univariate analysis ($p = 0.031$ and $p = 0.017$, respectively), but not in the multivariate analysis. Given the high prevalence of PFBMs in breast cancer patients it is possible that there is a synergistic effect between both variables. Finally, no significant survival differences were found in relation to brain tumor-related epilepsy, although due to the sample size, no comparative study was performed between patients with epilepsy before or after radiosurgical treatment. Therefore, further prospective studies are needed to evaluate this aspect of growing interest [15].

## 5. Conclusions

Stereotactic radiosurgery has been a significant advance in the approach to patients with brain metastases since it is a fast technique with less neurocognitive impact compared to conventional WBRT without losing anything in terms of local control of lesions or patient survival. Almost nine out of ten BMs present local control in the first neuroimaging control. The median survival of patients was 7.7 months, which was significantly longer in those

with a better baseline situation at the time of treatment (KPS) as well as those with a higher DS-GPA. Other variables such as the type of primary tumor, age, and the presence of metastases in the posterior fossa could also influence survival, although further studies with a larger sample size are needed to answer these questions.

**Author Contributions:** Conceptualization, J.M.S.-V., A.S.-B. and M.A.; methodology, J.M.S.-V. and E.M.-A.; validation, J.M.S.-V., J.S.-R. and P.P.E.-P.; formal analysis, J.M.S.-V. and M.A.-G.; investigation, J.M.S.-V. and A.S.-B.; writing—original draft preparation, J.M.S.-V.; writing—review and editing, J.M.S.-V. and J.A.P.-V.; supervision, J.S.-R., A.S.-B., M.A. and P.P.E.-P.; project administration, J.M.S.-V. and M.A. All authors have read and agreed to the published version of the manuscript.

**Funding:** This research received no external funding.

**Institutional Review Board Statement:** The study was conducted in accordance with the Declaration of Helsinki. Data collection and observational studies were approved by the Ethics Committee of the University Hospital Complex of Cartagena (Murcia, Spain). Protocol code TI 15/19 and date of approval (10 September 2014).

**Informed Consent Statement:** Patient consent was waived due to this is an observational and descriptive study, with retrospective review of medical records, where most of the patients are deceased at the time of the present work.

**Data Availability Statement:** Not applicable.

**Acknowledgments:** We thank Guadalupe Ruiz Merino, Methodological Support Unit (Area of Research, Innovation, and Development, International Biosanitary Projects Office (IBiPO). Murcia, Spain), for her support in some statistical tasks.

**Conflicts of Interest:** The authors declare no conflict of interest.

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
