# Peer review of "Volumetric Modulated Arc Therapy for Radiosurgery of Brain Metastases: A Single-Center Study"

_applsci, doi:10.3390/app131810097_

Round 1
Reviewer 1 Report
Accepted
Accepted
Author Response
- Comments and Suggestions for Authors: Accepted
- Comments on the Quality of English Language: Accepted
We would like to thank the referees for their efforts in reviewing the manuscript. We believe that the comments have improved the manuscript, making the messages more relevant and rigorous.
Reviewer 2 Report
This paper analyzes the key clinical and demographic aspects as well as radio-oncologic factors of a group of patients with BM who underwent treatment with Volumetric Modulated Arc Therapy for Radiosurgery. The study is retrospective in nature. (VMAT-RS).
The study employs suitable methodologies and thorough statistical testing. The results are presented using tables and graphs. This is among the initial studies that analyze a group of patients with BMs treated with radiosurgery using volumetric modulated arc therapy. The purpose is to provide an overall radiographic view of neuro-oncology patients by examining both the primary radio-oncologic and clinicodemographic factors.
Minor changes to improve the manuscript:
1. I noticed that in Figure 5, the font size of the axis legends and numbers is too small. Can you please enlarge it?
2. Add a reference to Eq. 1
3. Add a reference to "Eclipse 10.0 software" in line 139.
Author Response
This paper analyzes the key clinical and demographic aspects as well as radio-oncologic factors of a group of patients with BM who underwent treatment with Volumetric Modulated Arc Therapy for Radiosurgery. The study is retrospective in nature. (VMAT-RS).
The study employs suitable methodologies and thorough statistical testing. The results are presented using tables and graphs. This is among the initial studies that analyze a group of patients with BMs treated with radiosurgery using volumetric modulated arc therapy. The purpose is to provide an overall radiographic view of neuro-oncology patients by examining both the primary radio-oncologic and clinicodemographic factors.
Minor changes to improve the manuscript:
- I noticed that in Figure 5, the font size of the axis legends and numbers is too small. Can you please enlarge it?
- Add a reference to Eq. 1
- Add a reference to "Eclipse 10.0 software" in line 139.
(x) English language fine. No issues detected.
First of all, we would like to thank the reviewer for his time in reviewing our manuscript, as well as the comments regarding the review process. We hope that all questions are adequately answered.
First, we have improved some elements of Figures 2 and 5, improving the visibility of the legend, as well as modifying some of the titles (avoiding the use of acronyms) to make them easier to read for readers of the manuscript. Similarly, we have modified some elements of the caption, which could cause confusion to the reader.
New version:
Kaplan-Meier curves of patients treated with SRS/fSRS for: a) overall probability of survival. And survival according to b) KPS (cut-off 70), c) DS-GPA class, d) age grouping (cut-off 65 years), e) presence of posterior fossa brain metastasis, f) tumor type (lung vs breast), g) “brain tumor related epilepsy” and h) presence of extracranial metastasis.
Secondly, we have added the reference both to equation 1 on page 3 and to the eclipse software (line139), including in the latter case the name of the company.
Reviewer 3 Report
This study aimed to retrospectively assess the radio-oncologic and clinical-demographic aspects of brain metastase patients treated with Volumetric Modulated Arc Therapy for Radiosurgery (VMAT-RS). A cross-sectional observational study design was adopted, involving a retrospective review. This study underscores the viability of VMAT-RS as a treatment modality with a comparable overall survival rate to other radiosurgery techniques. While delving into other contributing variables such as primary tumor type, patient age, and PFBMs, the study acknowledges the need for further exploration to comprehensively unravel their impact on survival rates. The findings contribute to the broader understanding of VMAT-RS as a valuable treatment option for brain metastases, while simultaneously shedding light on areas that warrant further investigation.
Minor comments:
1) Axes, axes label & legends for figures 2 and 5 should be made more readable.
2) References need formatting; year and journal detail is missing for example #28

Author Response
This study aimed to retrospectively assess the radio-oncologic and clinical-demographic aspects of brain metastases patients treated with Volumetric Modulated Arc Therapy for Radiosurgery (VMAT-RS). A cross-sectional observational study design was adopted, involving a retrospective review. This study underscores the viability of VMAT-RS as a treatment modality with a comparable overall survival rate to other radiosurgery techniques. While delving into other contributing variables such as primary tumor type, patient age, and PFBMs, the study acknowledges the need for further exploration to comprehensively unravel their impact on survival rates. The findings contribute to the broader understanding of VMAT-RS as a valuable treatment option for brain metastases, while simultaneously shedding light on areas that warrant further investigation.
Minor comments:
- Axes, axes label & legends for figures 2 and 5 should be made more readable.
- References need formatting; year and journal detail is missing for example #28
(x) English language fine. No issues detected.
First of all, we would like to thank the reviewer for his time in reviewing our manuscript, as well as the comments regarding the review process. We hope that all questions are adequately answered.
First, we have improved some elements of Figures 2 and 5, improving the visibility of the legend, as well as modifying some of the titles (avoiding the use of acronyms) to make them easier to read for readers of the manuscript. Similarly, we have modified some elements of the caption, which could cause confusion to the reader.
New version:
Kaplan-Meier curves of patients treated with SRS/fSRS for: a) overall probability of survival. And survival according to b) KPS (cut-off 70), c) DS-GPA class, d) age grouping (cut-off 65 years), e) presence of posterior fossa brain metastasis, f) tumor type (lung vs breast), g) “brain tumor related epilepsy” and h) presence of extracranial metastasis.
Finally, the bibliographic references have been revised and corrected.